# Technique-Dependent Relationship between Local Ski Bending Curvature, Roll Angle and Radial Force in Alpine Skiing

**DOI:** 10.3390/s23083997

**Published:** 2023-04-14

**Authors:** Christoph Thorwartl, Andreas Tschepp, Michael Lasshofer, Helmut Holzer, Martin Zirkl, Matthias Hammer, Barbara Stadlober, Thomas Stöggl

**Affiliations:** 1Department of Sport and Exercise Science, University of Salzburg, Schlossallee 49, 5400 Hallein/Rif, Austria; michael.lasshofer@plus.ac.at (M.L.); thomas.stoeggl@plus.ac.at (T.S.); 2Joanneum Research Forschungsgesellschaft mbH—MATERIALS, Franz-Pichler-Straße 30, 8160 Weiz, Austria; andreas.tschepp@joanneum.at (A.T.); martin.zirkl@joanneum.at (M.Z.); matthias.hammer@joanneum.at (M.H.); barbara.stadlober@joanneum.at (B.S.); 3Atomic Austria GmbH, Atomic Strasse 1, 5541 Altenmarkt, Austria; helmut.holzer@atomic.com; 4Red Bull Athlete Performance Center, Brunnbachweg 71, 5303 Thalgau, Austria

**Keywords:** bending sensors, flexion, PyzoFlex, ski bending, ski deflection

## Abstract

Skiing technique, and performance are impacted by the interplay between ski and snow. The resulting deformation characteristics of the ski, both temporally and segmentally, are indicative of the unique multi-faceted nature of this process. Recently, a PyzoFlex^®^ ski prototype was presented for measuring the local ski curvature ($w^{\prime \prime }$), demonstrating high reliability and validity. The value of $w^{\prime \prime }$ increases as a result of enlargement of the roll angle (RA) and the radial force (RF) and consequently minimizes the radius of the turn, preventing skidding. This study aims to analyze segmental $w^{\prime \prime }$ differences along the ski, as well as to investigate the relationship among segmental $w^{\prime \prime }$, RA, and RF for both the inner and outer skis and for different skiing techniques (carving and parallel ski steering). A skier performed 24 carving and 24 parallel ski steering turns, during which a sensor insole was placed in the boot to determine RA and RF, and six PyzoFlex^®^ sensors were used to measure the $w^{\prime \prime }$ progression along the left ski ($w_{1-6}^{\prime \prime }$). All data were time normalized over a left-right turn combination. Correlation analysis using Pearson’s correlation coefficient (r) was conducted on the mean values of RA, RF, and segmental $w_{1-6}^{\prime \prime }$ for different turn phases [initiation, center of mass direction change I (COM DC I), center of mass direction change II (COM DC II), completion]. The results of the study indicate that, regardless of the skiing technique, the correlation between the two rear sensors ($L_{2}$ vs. $L_{3}$) and the three front sensors ($L_{4}$ vs. $L_{5}$, $L_{4}$ vs. $L_{6}$, $L_{5}$ vs. $L_{6}$) was mostly high (r > 0.50) to very high (r > 0.70). During carving turns, the correlation between $w^{\prime \prime }$ of the rear ($w_{1-3}^{\prime \prime }$) and that of front sensors ($w_{4-6}^{\prime \prime }$) of the outer ski was low (ranging between −0.21 and 0.22) with the exception of high correlations during COM DC II (r = 0.51–0.54). In contrast, for parallel ski steering, the r between the $w^{\prime \prime }$ of the front and rear sensors was mostly high to very high, especially for COM DC I and II (r = 0.48–0.85). Further, a high to very high correlation (r ranging between 0.55 and 0.83) among RF, RA, and $w^{\prime \prime }$ of the two sensors located behind the binding ($w_{2}^{\prime \prime },w_{3}^{\prime \prime }$) in COM DC I and II for the outer ski during carving was found. However, the values of r were low to moderate (r = 0.04–0.47) during parallel ski steering. It can be concluded that homogeneous ski deflection along the ski is an oversimplified picture, as the $w^{\prime \prime }$ pattern differs not only temporally but also segmentally, depending on the employed technique and turn phase. In carving, the rear segment of the outer ski is considered to have a pivotal role for creating a clean and precise turn on the edge.

## 1. Introduction

### 1.1. Design and Material Properties of Alpine Skis

In alpine skiing, not only human factors but also equipment properties influence the skiing style. The parabolic shape of the ski is an essential design characteristic and can be classified by sidecut radius ($R_{SC}$). A carving ski has a progressive sidecut shape; therefore, the ski width increases toward the shovel and the tail, with the narrowest part underneath the binding. Viewed from above, the $R_{SC}$ is determined by the circle resulting from the shovel, the tail, and the point of the narrowest width in the middle of the ski, and it defines the geometrical turn radius [1,2,3]. Furthermore, the length of the ski, or more precisely the length of the ski’s running surface that is in contact with the snow, has a significant effect on the ski’s turning behavior, whereby shorter skis are easier to turn on the snow [1]. When a carving ski is placed on a flat surface, both ends (shovel and tail) touch the ground, while the middle part is not in contact with the ground, creating an arc shape called a camber [1,2]. The camber is designed to effectively distribute the pressure over the tip and tail of the ski. In addition, the bending and torsional stiffness also affect the turning behavior of the ski and determine how aggressively the ski tip and tail interact with the snow surface when the ski is edged and loaded [4,5,6]. However, proper balance among ski length and width, $R_{SC}$, camber, bending stiffness, and torsional stiffness creates the performance characteristics of the ski.

### 1.2. Skiing Techniques

Besides non-parallel skiing styles, two main skiing styles are generally recognized: carving and parallel ski steering, also known as skidding. Carving is the more dynamic parallel skiing style and is generally used by experienced and highly trained skiers. It refers to the technique in which the tip of the ski forms a groove in the snow that the rest of the ski follows, resulting in a self-steering effect [2,5]. In contrast, during a parallel ski steering turn, a point along the edge does not follow the path of the proceeding ski but slides laterally across the slope [4,7,8,9]. For practical reasons, the majority of the literature refers to parallel ski steering or carving, but these processes are not dichotomous; rather, both can occur simultaneously in different segments along the ski, depending on the so-called angle of attack [4,10]. To ensure that the angle of attack is as small as possible, the vector of the skier’s translational velocity must always point in the direction of the orientation of the ski [4,7]. Turns with a high proportion of carving are only possible if the ski deflects over the entire length so that all edge points follow the same trajectory line closely.

### 1.3. The Science of Ski Turns

The term “turn radius” is commonly used to describe the path of a turn. The equipment-related $R_{SC}$ does not correspond to the actual turn radius, but it does affect the magnitude of ski deflection when the ski is edged and loaded [1,2,11]. If a ski is inclined at a specific edge angle (θ) and pressed against a firm and flat surface, causing it to bend and make contact with the snow, the resulting deformation can still be characterized as an arc with an edge curvature radius $R_{e}$ [12].
(1)$$R_{e}=R_{SC}\cdot cos(\theta )$$

Equation (1) presents a generalized representation of the ski’s deformation, as the ski’s bending characteristics are not homogeneous but rather exhibit variations in segmental curvature along its length [10,13]. Thus, to represent the inhomogeneous deflection pattern, it would be an oversimplification to fit a circle with a constant radius to the ski’s bending line. A more accurate representation would require multiple circles with varying local ski curvature ($w^{\prime \prime }$).

However, to allow the ski to deflect, the ski must be edged, and a sufficiently large radial force (RF) must be applied (Figure 1a), leading to the deduction that the greater θ, the more RF must be realized, and the more the ski must be deflected to reduce R so that the entire edge remains in contact with the snow (Figure 1a,b) [2,5]. It should be noted that, in the subsequent sections, the term “edge angle” is not used. Instead, the term “roll angle” (RA), which is equivalent to θ on a flat terrain and independent of slope inclination, is employed. Although “lean angle” is used in other literature, it is often associated with the angle at which the skier’s body leans, rather than the angle of the ski’s edge. Therefore, we recommend using the more precise term RA to describe the angle of the ski’s edge, avoiding confusion and accurately conveying the intended meaning. It is assumed that, in soft snow conditions, smaller RA values are sufficient to produce the same turn radius as in hard conditions, as the ski penetrates the snow in the middle ski segment easier, to reduce the $w^{\prime \prime }$ of the ski (Figure 1c). It should be noted that the torsional deformation of the ski, which is more pronounced in hard slope conditions, leads to a reduction in the local RA. Segmental differences in RA due to torsional effects also reduce the local bending radii along the ski.

### 1.4. State of Current Knowledge and Research Gap

The use of plantar pressure distribution sensors to measure load distribution on skis is a commonly employed technique in the field [14,15,16,17,18,19,20]. Studies have shown that the ability to generate higher RF is associated with improved performance [21] because a greater proportion of the net snow reaction forces are oriented radially, indicating that the turn was executed with a larger proportion of carving [21]. Furthermore, it has been shown that θ also is a key factor in determining carving performance, as higher θ results in a smaller R [22]. Research in competitive environments has indicated that a larger proportion of carving and a smaller turn radius lead to enhanced performance [23,24,25]. Studies of the detection of θ or RA have gained increasing attention in recent years [26,27,28,29].

However, the impact of local ski $w^{\prime \prime }$ on carving performance has not yet been investigated. To measure the characteristic $w^{\prime \prime }$ progression along the ski, there are two options: a strain gauge-based prototype [30] or a PyzoFlex^®^ technology-based prototype [13]. Laboratory investigations have demonstrated the reliability and validity of the novel device in providing accurate measurements of $w^{\prime \prime }$ progression in both quasi-static [13] and dynamic conditions [10,31]. Furthermore, plausible data have already been detected during proof-of-concept field measurements while skiing but with smaller data sets and partly without reference systems [32,33,34].

### 1.5. Aim and Research Question

The primary objective of this study is to examine: (i) the correlation between the $w^{\prime \prime }$ sensors ($L_{1}\;to\;L_{6}$) along the ski to infer possible segmental curvature differences; and (ii) the relationship among RA, RF, and segmental $w^{\prime \prime }$. A distinction is made in each case between the inner and outer skis and between parallel ski steering and carving.

## 2. Materials and Methods

### 2.1. Experimental Setup and Measurement Equipment

One skier (A-level ski instructor) performed 48 carving turns (24 short and 24 long radii each) and 48 parallel ski steering turns (24 short and 24 long radii each) over six runs. The instrumented ski (Atomic Redster G7) utilized in the study has a length of 1.82 m and an $R_{SC}$ of 19.6 m. The study was conducted on a homogeneous slope with compact snow conditions. For the long radii turns, a medium slope steepness was chosen, and for the short radii turns, a range of medium to great slope steepness was chosen. To identify the turn switch points (TSPs), simultaneous recordings were made using a GoPro Hero 4 Black camera (GoPro, Inc., San Mateo, CA, USA) on the body and a follow-up camera (EOS RP, Canon, Melville, NY, USA) both operating at a frame rate of 50 fps. The TSP in skiing refers to the moment when the ski is placed flat on and is transitioned from one edge to the other, leading to a change in the direction of motion [26]. An OpenGo sensor insole (Moticon GmbH, Munich, Germany) was placed in the ski boots to provide both, plantar pressure and inertial measurement unit (IMU) data at a sampling rate of 100 Hz. All Moticon OpenGo data recorded during the measurement were filtered using a second-order Butterworth bandpass filter with low and high cut-off frequencies of 0.2 and 6 Hz, respectively, in MATLAB (R2018B, MathWorks, Natick, MA, USA). RA was calculated by integrating the filtered rotational velocity ($\omega_{y}$) around the y-axis (roll axis) over time (t), as shown in Equation (2).
(2)$$RA=\int \omega_{y}\cdot dt$$

RA is assumed to be zero at the TSPs; therefore, these points serve as anchor points for the IMU-based RA calculation. A customized zero-update algorithm was implemented using MATLAB, which consistently sets the RA to zero at the TSP and corrects the linear drift to the next TSP according to the procedure of Snyder and colleagues [28]. The Moticon system enables the measurement of the foot forces obtained by the pressure insoles, as well as the evaluation of the center of pressure in both the medial-lateral (*x*) and anterior-posterior (*y*) directions. RF was calculated by multiplying the foot force by the sine of |RA|. It should be noted that the effective foot force between the ski and binding system exceeds the estimate from the sensor insoles due to loss of force components through the boot shell and cuff [16,17,19]. However, the force data obtained from the sensor insoles were deemed sufficient for the purposes of this study, which primarily focuses on analyzing the correlation coefficient, the main metric of the analysis, rather than interpreting the absolute force values.

The PyzoFlex^®^ technology-based ski prototype (Joanneum Research Forschungsgesellschaft m.b.H, Weiz, Austria, sampling rate = 215 Hz) previously presented in another paper was used to measure the segment-wise $w^{\prime \prime }$ [10]. Due to their intrinsic piezoelectric and pyroelectric nature, PyzoFlex^®^ sensors produce a charge Q(t), which is converted into a proportional output voltage $u_{a}(t)$. A segment-wise curvature $w_{i}^{\prime \prime }$ was calculated from each sensor element i as a function of the respective sensor voltage $u_{a,i}$ using the two-point calibration values ($k_{i}$ and $d_{i}$) from Equation (3).
(3)$$w_{i}^{\prime \prime }\left(u_{a,i}\right)=k_{i}\cdot u_{a,i}+d_{i}$$

The calibration values were determined on a bending machine with a high-precision laser measurement system using a self-developed curvature model [26]. Before calibration, all data were filtered and corrected for linear drift using MATLAB. Specifically, a second-order Butterworth high-pass filter with a cut-off frequency of 0.2 Hz was applied to remove temperature-related fluctuations in the raw data caused by the pyroelectric effect. The sensor data, recorded with a data acquisition (DAQ) device, were stored directly on a built-in SD card and transferred to a laptop via WLAN after the measurement. The DAQ device was also coupled with a synchronization unit (“sync tool”), which allowed for starting and stopping the measurements and synchronizing the sensor data with the video recording by switching an LED that could be filmed by an external camera. The corresponding trigger points (i.e., switching times of the LED) were stored in the database together with the sensor data and enabled post-synchronization with the video stream. To synchronize the PyzoFlex and video system with the Moticon OpenGo system, an additional jump was performed at the beginning and end of each trial. The landing of the jump was used as a synchronization event since the foot force measured with the pressure sole showed a clear peak at this point.

The experimental setup—including: (1) body cam; (2) follow cam; (3) PyzoFlex^®^ ski prototype with sync tool and DAQ device; and (4) Moticon sensor insoles—is shown in Figure 2.

### 2.2. Data Processing and Statistical Analysis

To provide a qualitative description, all data were time-normalized over a left-right turn combination. The mean RA ($\bar{RA}$), mean RF ($\bar{RF}$), and mean segmental $w^{\prime \prime }$ ($\bar{{w_{i}}^{\prime \prime }}$) were calculated for each of the six sensor elements i over all carving and parallel ski steering turns, with the corresponding standard error (SE) reported. Additionally, the maximum values ($\bar{{RA}_{MAX}}$ or $\bar{{RA}_{MIN}}$, $\bar{RF_{MAX}}$, $\bar{w_{i,MAX}^{\prime \prime }}$) for both the inner and outer skis were determined from these mean curves. Each left-right turn combination was divided into four different phases—initiation, center of mass direction change I (COM DC I), center of mass direction change II (COM DC II), and completion—for both the inner ski (0% to 12.5%, 12.5% to 25%, 25% to 42.5% and, 42.5 to 50%) and the outer ski (50% to 62.5%, 62.5% to 75%, 75% to 92.5%, and 92.5 to 100%) [35]. For correlation analysis, the mean of RA, RF, and segmental $w_{1-6}^{\prime \prime }$ were calculated in the corresponding sections. The relationship between the 6 $w^{\prime \prime }$ sensors and the correlation between segmental $w_{1-6}^{\prime \prime }$, RA, and RF were analyzed for both the inner and outer skis and separated for the carving and parallel ski steering techniques. The magnitude of agreement was assessed using Pearson’s correlation coefficient (r) with thresholds of 0.1, 0.3, 0.5, and 0.7 for small, moderate, large, and very large correlations, respectively [36]. In order to improve the readability of the manuscript, a list of corresponding abbreviations has been added to the Abbreviations section.

## 3. Results

### 3.1. Descriptive Report

The $\bar{RA}$ ± SE, $\bar{RF}$ ± SE, and $\bar{{w_{i}}^{\prime \prime }}$ ± SE of left-right turn combinations, normalized by time, are shown for carving in Figure 3a and for parallel ski steering in Figure 3b. At 0% and 100%, the TSP marks the transition from a right to a left turn and, at 50%, from a left to a right turn.

The highest $\bar{RA}$ and $\bar{RF}$ are observed in COM DC II for both the inner and outer skis. During carving, the max $\bar{RF}$ is around 450–500 N greater than during parallel ski steering; this maximum applies for both the inner ski ($\bar{RF_{MAX}}$ = 517 N vs. $\bar{RF_{MAX}}$ = 56 N) and the outer ski ($\bar{RF_{MAX}}$= 1111 N vs. $\bar{RF_{MAX}}$ = 556 N). In carving, the max $\bar{RA}$ is higher on the outer ski ($\bar{RA_{MAX}}$ = 57°) compared to the inner ski ($\bar{RA_{MIN}}$ = 53°). Conversely, during parallel ski steering, the inner ski exhibits a greater degree of edging ($\bar{RA_{MAX}}$ = 45°) than the outer ski ($\bar{RA_{MAX}}$ = 42°). Among the sensor elements i, the one located behind the binding system ($L_{3}$) shows the greatest $w_{i,MAX}^{\prime \prime }$ of 0.26 m^−1^ in carving, followed by the neighboring element $L_{2}$ ($w_{2,MAX}^{\prime \prime }$= 0.22 m^−1^) and the sensor $L_{4}$ ($w_{4,MAX}^{\prime \prime }$ = 0.20 m^−1^) in front of the binding. Figure 4 shows that, during carving, $\bar{w_{i,MAX}^{\prime \prime }}$ moves progressively over time from the front sensor $L_{6}$ to the rear sensor $L_{1}$. In contrast, no sequential course of $\bar{w_{i,MAX}^{\prime \prime }}$ is visible in parallel ski steering. The time-normalized center of pressure progression in the medial-lateral (x) and anterior-posterior (y) directions, which was not included in the correlation analysis but is relevant for the discussion, is shown in Figure A1 of the Appendix A.

### 3.2. Correlation between Curvature Sensors

Table 1 shows the bivariate correlations between the $w^{\prime \prime }$ of the sensor segments $L_{1}$ to $L_{6}$ for different turn phases and techniques. It was found that, for carving, the front sensors $(L_{4}$ vs. $L_{5}$, $L_{4}$ vs. $L_{6}$, $L_{5}$ vs. $L_{6}$) have a very high correlation with each other across all turn phases (r = 0.70–0.98). The correlation is comparatively high for parallel ski steering, whereby lower correlations are partially evident for the inner ski (r= 0.38–0.93). The rear sensors $L_{2}$ and $L_{3}$ are very highly correlated with each other (r = 0.70–0.98) over all techniques, except for the inner ski during the initiation phase of parallel ski steering (r = 0.64). The rear sensor $L_{1}$ shows only low to moderate correlations with the other sensors during carving (r= −0.22–0.48), apart from a relatively high correlation with $L_{2}$ in COM DC II (r = 0.56) for the outer ski. The correlation between the $w^{\prime \prime }$ of the rear sensors ($L_{1}$ to $L_{3}$) and the front sensors ($L_{4}$ to $L_{6}$) of the outer ski during carving is low, with a correlation coefficient ranging from −0.21 to 0.22, with the exception of a high correlation (r > 0.50) observed during the COM DC II phase. Further analysis revealed that, for the inner ski during carving, a high correlation between two sensors behind the binding ($L_{2}$ and $L_{3}$) and the front sensors ($L_{4}$ to $L_{6}$) was observed only in the COM DC II phase (r = 0.50–0.57). For the remaining three turn phases in carving, low to moderate or even high negative correlations (r = 0.35–0.63) between the front and rear sensors can be observed. In contrast to the findings during carving, parallel ski steering revealed mostly high to very high correlations between the front and rear sensors, particularly for the outer ski, but not during the initiation phase (r = −0.35 −0.23). To provide a more in-depth look at the temporally different correlations of the turn phases, the authors have included tables in the Appendix B for both carving (Table A1 and Table A2) and parallel ski steering (Table A3 and Table A4).

### 3.3. Correlation between RF, RA and Segmental w″

The r values among RA, RF, and ${w_{i}}^{\prime \prime }$ for various turn phases (initiation, COM DC I, COM DC II, and completion) are illustrated in Figure 5a for carving and in Figure 5b for parallel ski steering. Data from the red-highlighted inner ski were analyzed within the range of 0–50% of the turn, while data from the red-highlighted outer ski were analyzed within the range of 50–100% of the turn. To make the plot in Figure 5a clearer, only the mean value of r is shown for the front sensors ($L_{4}$ to $L_{6}$), which showed high similarity. To conduct a comprehensive evaluation of the data, the authors have added supplementary tables in the Appendix C for both carving (Table A5 and Table A6) and parallel ski steering (Table A7 and Table A8).

Excluding the initiation phase (r = 0.53), a very high, positive correlation between RA and RF (r = 0.79–0.83) was observed for the outer ski during carving. In contrast, during parallel ski steering, moderately negative to moderately positive correlations (r = −0.33–0.45) between RA and RF were noted. For the inner ski, the r values demonstrated a variation from low to high (r = −0.01–64) across both carving and parallel ski steering techniques. It is notable that, with the exception of a moderate correlation observed in the COM DC II phase (r = 0.34), the front sensors ($L_{4}$ to $L_{6}$) demonstrated primarily low to negative high relationships between RA and RF versus $w_{4-6}^{\prime \prime }$ during carving for both the inner and outer skis. In contrast, the two rear sensors ($L_{2}$ and $L_{3}$) exhibited exclusively positive r during carving, specifically for the outer ski. In the COM DC II phase, a very high r ranging between 0.73 and 0.80 was observed for $w_{2}^{\prime \prime }$ and $w_{3}^{\prime \prime }$ vs. RA and RF, while in the other phases, moderate to very high correlation coefficients (r = 0.30–0.74) were visible. During parallel ski steering of the outer ski, there were two high correlations between ${w_{3}}^{\prime \prime }$ and the RF (r = 0.54) in the completion phase and between $w_{6}^{\prime \prime }$ and the RA (r = 0.54) in the COM DC I phase; all other correlation variants among RA, RF, and $w_{1-6}^{\prime \prime }$ ranged between moderately negative and moderately positive (r = −0.33–0.46) (Table A7). For the inner ski, the correlation coefficients varied depending on the sensor position, turn phase, and skiing technique.

## 4. Discussion

The objective of this study was to analyze the difference in segmental $w^{\prime \prime }$ along the ski and to investigate the relationship among $w_{1-6}^{\prime \prime }$, RA, and RF for both the inner and outer skis, as well as for different techniques (carving and parallel ski steering).

The results indicate that the $w^{\prime \prime }$ characteristic differs both temporally and segmentally with respect to the inner and outer skis and the performed technique. The relationship between the two rear sensors ($L_{2}$ vs. $L_{3}$) and within the three front sensors ($L_{4}$ vs. $L_{5}$, $L_{4}$ vs. $L_{6}$, $L_{5}$ vs. $L_{6}$) was found to be mostly high or very high, regardless of the skiing technique. However, it is remarkable that, especially in carving, there are only low relationships between the front and rear $w^{\prime \prime }$ of the ski, except for the COM DC II phase. This finding reflects the ski’s inhomogeneous $w^{\prime \prime }$ pattern and the corresponding decoupling of front and rear segment bending caused by the binding [37]. The prediction that the w″ of the ski increases with larger RA and larger RF is predominantly valid for the rear sensors $L_{2}$ and $L_{3}$ of the outer ski during carving in the COM DC I and COM DC II phases. In contrast, the $w^{\prime \prime }$ of the three front sensors ($L_{4}$ to $L_{6}$) show significantly lower or even high negative correlations with RA and RF.

The force data collected in our study align with the results of previous research [14,15,16,17,18,19,38,39], indicating a substantial discrepancy in the measured force between the outer and inner skis, with the force on the outer ski being greater. Our measurements indicate a maximum RF of the outer ski of approximately 1.5 times the subject’s body weight, a result closely corresponding with the findings of Cross et al. (1.4 × body weight) [21]. The maximum RA measurement during carving was approximately 10° smaller than the θ of high-performance athletes (members of the Norwegian national team) from a previous study [4]. This discrepancy may be attributed to our only measuring the slope-independent RA and not the θ (Figure 1) and our skier not reaching the dynamic limit with the prototype during testing. In carving, both $\bar{RF_{MAX}}$ and $\bar{{RA}_{MAX}}$ are larger compared to during parallel ski steering, conforming to the prior literature [4,21]. The $w^{\prime \prime }$ values collected in the field test were consistent with the skiing-like bending deformation observed on a bending robot, with recorded sensor maximums ranging from 0.15 m^−1^ to 0.38 m^−1^ [10]. Moreover, the rear segment of the ski exhibited greater deformation, a finding in line with previous studies involving lab data from a bending machine [13], an oscillation machine [31], a programmable bending robot [10], and field results from a strain gauge-based prototype [30,40,41].

In perfect carving, according to the definition, every point along the ski’s edge follows the same path, resulting in a minimal track width, whereas in parallel ski steering, each point along the edge follows a distinct trajectory (with angle of attack≫0) [4,7,8,9]. Therefore, carving requires a precise alignment between the ski’s segment-wise orientation and the vector of its local translational velocity (${\overset{⃑}{v}}_{i}$), resulting in a minimal local angle of attack ($\varphi_{i}$) across all ski segments i (Figure 6a). During real-world carving conditions, our findings indicate that the effective carving segment is mostly concentrated to the rear part of the ski (Figure 6b), where the highest $w^{\prime \prime }$ is observed, and there is a strong correlation with RF and RA, in contrast to parallel ski steering, as visualized in Figure 6c. This finding is consistent with previous studies, which have demonstrated that, during an advanced carving stage, the angle of attack in the ski’s forebody exhibits a slight increase in $\varphi_{i}$, indicating that this segment of the ski was still machining new snow [2]. However, the shovel’s function is to create a groove for the rear part of the ski to glide in [2,41], providing an explanation for the earlier temporal activation of the front sensors during carving, in contrast to non-progressive behavior in parallel ski steering (Figure 4). The earlier activation of the front sensors ($L_{4}$ to $L_{6}$) during carving can be attributed to the center of pressure of the outer ski, which is located anteriorly during the initiation phase and gradually shifts toward the back of the foot as the turn progresses in carving (Figure A1). Furthermore, the statement that the $R_{e}$ is identical to the local $w^{\prime \prime }$ radius of the ski trajectory, as derived from Equation (1), is an oversimplification [12]. The bending of a ski cannot be accurately represented by a single circle with a constant radius; thus, the local $w^{\prime \prime }$ radius is an inadequate descriptor of the ski’s turn radius. A prototype for measuring ski bending using strain gauges has indicated that the effective bending radius of the ski during carving is approximately twice as large as the true turn radius [42]. The non-deducibility of the ski deflection to the true turn radius can also be supported in this work, as shown by the peak bending ($w_{3,MAX}^{\prime \prime })$ of 0.26 m^−1^ (Figure 3a). This outcome corresponds to a radius of approximately 3.85 m, which is considerably smaller than the calculated peak $R_{e}$ of 10.80 m derived from Equation (1) using $R_{SC}$ = 19.6 m and $\bar{RA_{MAX}}$ = 57°. However, in a previous study conducted by another research team, it was found that the radius calculated from deflection measurements showed better agreement with the center of mass radii obtained using a differential Global Navigation Satellite System [43]. Nevertheless, it is important to note that no segment-wise analysis was conducted to obtain a more detailed picture.

There are limitations to this study. While the RA changes along the length of the ski, the correlation analysis only considers the global RA under the ski boot and not the local RA at the corresponding sensor segments. Additionally, it should be noted that no torsional model is currently implemented in the PyzoFlex^®^-based bending calculation. Therefore, future studies should consider incorporating a torsional model to gain a more complete understanding of the interaction between local RA and deformation characteristics along the ski, including both local torsional angle and ${w_{i}}^{\prime \prime }$. Furthermore, the use of sensor insoles may not be optimal since previous research has shown that the compressive force is underestimated compared to a force platform during skiing [16,17,19]. However, the sensor insoles used in literature (e.g., PEDAR) were not calibrated on site. In contrast, in the current study, on-field calibration was conducted before each run using a specialized mechanism developed by the manufacturer to adjust the sensor unit to the subject’s body weight. The Moticon OpenGo sensor insole used in this study also features an automatic zeroing function that eliminates pre-activations (e.g., from tightly fastened ski boots) or temperature-related dependencies. Therefore, making a direct comparison between the pressure data obtained in this study and the literature data is only partially feasible. Further studies are needed to determine whether these mechanisms have improved the previously mentioned underestimation of force values. Despite limitations with the force data obtained from the sensor insoles, they are deemed sufficient for the purposes of this study, which primarily focuses on analyzing the correlation coefficient as the main metric, rather than interpreting the absolute force values. The percentage breakdown of initiation, COM DC I, COM DC II, and completion varies depending on the turn radius [35]. The authors chose to use the same percentage breakdown for both long and short radii to enable a consistent interpretation of the temporal $w^{\prime \prime }$ variation. For the IMU-based calculation of the RA, a zero-update routine was applied, in which the TSPs served as anchor points. This process enabled a linear drift correction from turn to turn. However, for the PyzoFlex^®^ signal, an analogue algorithm is currently not feasible, as the $w^{\prime \prime }$ at the TSP is not necessarily zero [10]. In the future, fusion with other sensors or sensor systems needs to be conducted to identify potential anchor points for segment-wise drift correction of the PyzoFlex^®^ data to further improve signal quality. Additionally, it is important to acknowledge that the impact of temperature on the $w^{\prime \prime }$ calculation is not yet fully understood due to the pyroelectric nature of the sensors. The prevalent assumption is that temperature changes with low frequencies can be eliminated using a high-pass filter. However, to determine the real impact of temperature and whether a temperature gradient should be incorporated into the model, bending measurements must be conducted in a cooling chamber.

Reducing the number of sensors required for a commercial application is often practical, as it can reduce the cost and complexity of the system. For instance, in a skiing performance monitoring system, it may be advantageous to only collect data from sensors located in areas that are most discriminative for performance. In particular, $L_{3}$ provides performance-relevant insights in this study, as it differentiates between carving and parallel ski steering. Further, $L_{5}$ is redundant and can be eliminated from the data collection process, as it is highly correlated with $L_{4}$ and $L_{6}$ (Table 1) and does not provide any unique or sophisticated insights. Despite the high r between Sensor $L_{2}$ and Sensor $L_{3}$, it is advisable to keep monitoring $L_{2}$ because of its importance in the temporal $w^{\prime \prime }$ sequence (Figure 4). However, despite the promising initial findings, further research is needed to determine whether the results of this approach can be generalized to a broader population.

## 5. Conclusions

The findings of the presented work provide insight into the unique patterns of segmental deflection behavior for different alpine skiing techniques and turn phases, and they link $w^{\prime \prime }$ to already well-researched performance-relevant RF and RA (or θ) data. The deflection of skis, which results from complex interactions between the ski and the snow, plays a significant role in skiing performance and turn quality. However, homogenous ski deflection along its length is an oversimplification. The $w^{\prime \prime }$ pattern varies not only temporally but also segmentally, depending on the technique employed and the phase of the turn. According to the research, a clean and accurate turn on the edge during carving relies heavily on the rear segment of the outer ski. To the best of our knowledge, there is currently no commercial application available for measuring ski deflection during skiing. PyzoFlex^®^ technology offers a promising avenue for the development of intelligent ski equipment, as it provides insight into segmental and temporal curvature-pattern-based performance analysis. The practical importance of this data makes it valuable for a range of applications, including real-time feedback systems, customization of ski products, injury analysis and prevention, and development of indoor testing methodologies for assessing the quality of the ski to reduce time-consuming on-field ski tests. To enhance ski design and achieve better equipment-athlete compatibility, future experimentation will involve testing different ski designs that vary in terms of bending and torsional stiffness, as well as incorporate left and right asymmetric designs. Furthermore, to enable more generalized performance statements for both the athletes and the equipment, multiple athletes will be included in these tests.

## Figures and Tables

**Figure 1 sensors-23-03997-f001:**
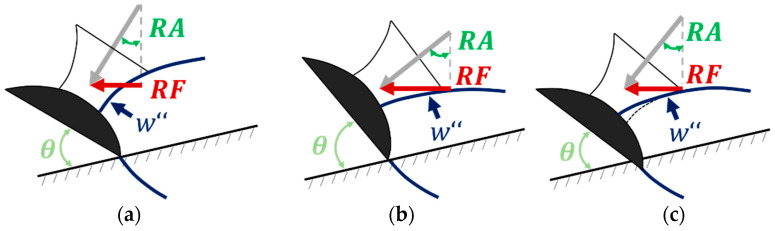
Relationship among radial force (RF), curvature ($w^{\prime \prime }$), edge angle (θ), and roll angle (RA) in (**a**,**b**) compact and (**c**) soft snow conditions based on LeMaster [5]. Due to the soft snow surface in (**c**), the middle segment of the ski is able to penetrate the snow, allowing it to cut the same radii as in (**b**).

**Figure 2 sensors-23-03997-f002:**
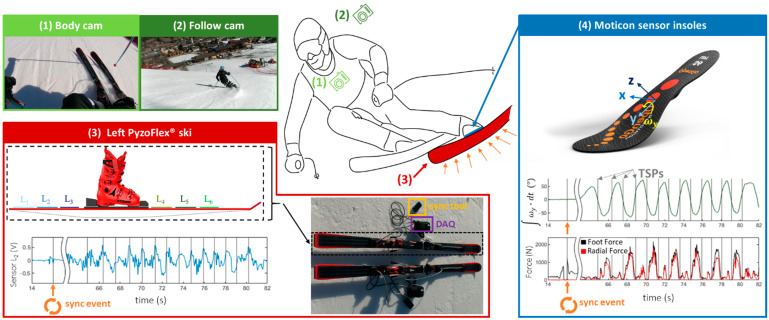
Experimental setup with: (**1**) body cam; (**2**) follow cam; (**3**) PyzoFlex^®^ ski prototype with data acquisition (DAQ) device, sync tool, and sensors $L_{1}$ to $L_{6}$; and (**4**) Moticon sensor insoles with the coordinate system (x,y,z) and the rotational velocity ($\omega_{y}$) around the y-axis for the roll angle (RA) calculation. The vertical gray lines in the diagrams correspond to the video-based turn switch points (TSP).

**Figure 3 sensors-23-03997-f003:**
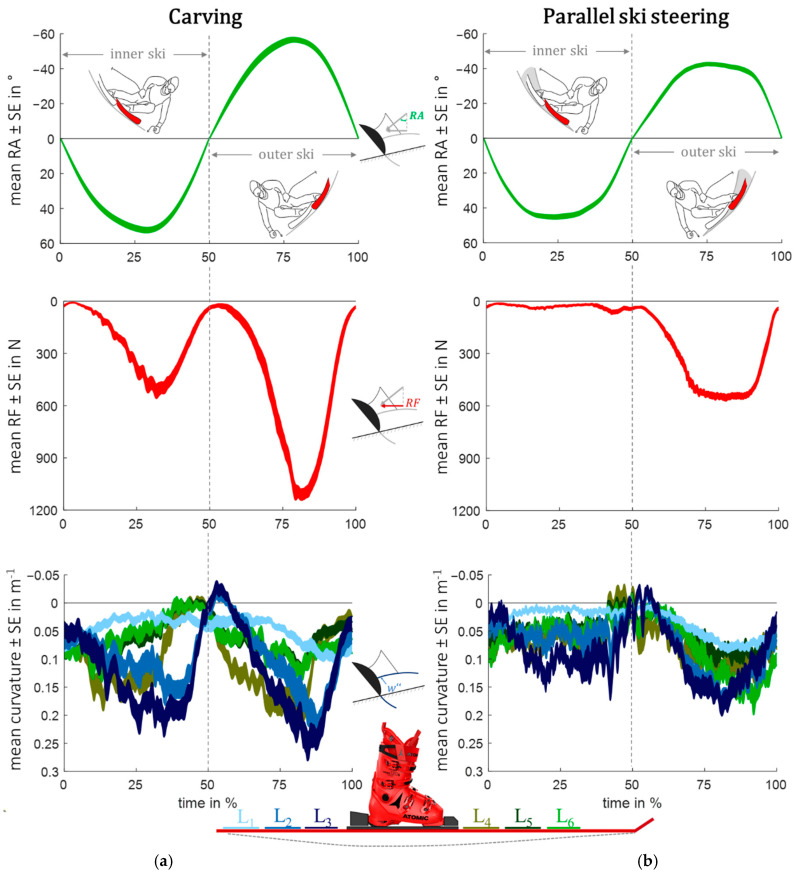
Mean +/− standard error (SE) of time-normalized roll angle (RA), radial force (RF), and segmental curvature (${w_{i}}^{\prime \prime }$) in carving turns (**a**) and parallel ski steering turns (**b**). In the lower graphs, the rear sensors are depicted in blue tones while the front sensors are represented in green tones, consistent with the ski illustration.

**Figure 4 sensors-23-03997-f004:**
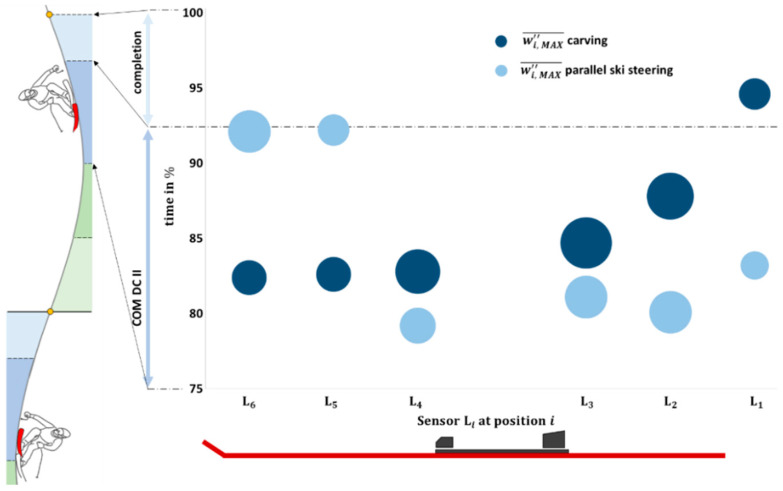
The segmental and temporal curvature characteristics of skiing differ between carving (dark blue) and parallel ski steering (light blue). During carving, the peak curvature ($\bar{w_{i,MAX}^{\prime \prime }}$) progressively moves in time from the front sensor $L_{6}$, over sensor positions i to the rear sensor $L_{1}$, with the highest curvature at $L_{3}$. On the other hand, no sequential course of $\bar{w_{i,MAX}^{\prime \prime }}$ is recognizable in parallel ski steering.

**Figure 5 sensors-23-03997-f005:**
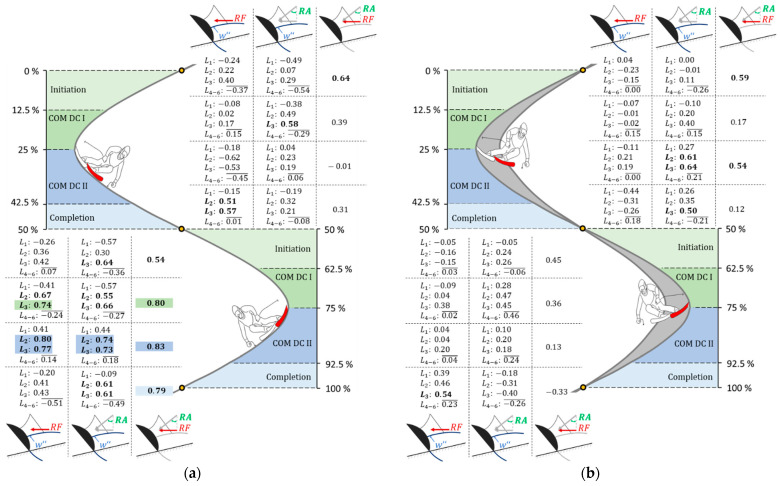
Pearson’s correlation coefficient among radial force (RF), roll angle (RA), and segmental curvature ($w^{\prime \prime }$) for sensor positions $L_{1}$ to $L_{6}$ at different turn phases (initiation; COM DC I: center of mass direction change one; COM DC II: center of mass direction change two; completion). A further distinction is made between (**a**) carving and (**b**) parallel ski steering, where the inner ski is shown at the top and the outer ski at the bottom. High, positive correlations (r > 0.50) are highlighted in bold, and very high correlations (r > 0.70) are highlighted with the corresponding color of the turn phase.

**Figure 6 sensors-23-03997-f006:**
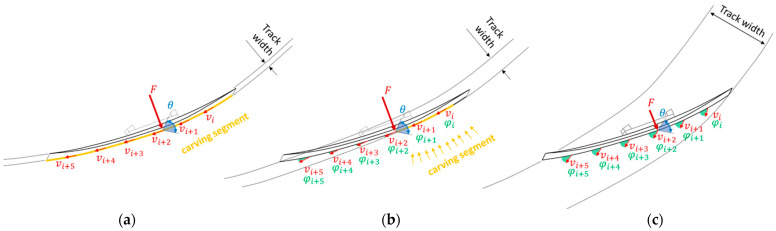
(**a**) Perfect carving according to definition, (**b**) carving under real-world conditions, and (**c**) parallel ski steering. Both the acting forces (F) and the edge angle (θ) are greater during carving than in parallel ski steering. In perfect carving, each point along the edge follows the other (local angle of attack ($\varphi_{i}$) between ski orientations, and the local translational velocity ($v_{i}$) along the ski is zero over all segments i), whereas in in parallel ski steering, all edge points follow their own trajectory ($\varphi_{i}\gg 0)$. Under real-world carving conditions, the effective carving segment is predominantly limited to the ski’s rear part, as indicated by the findings. The figure is based on Reid et al. [4].

**Table 1 sensors-23-03997-t001:** Pearson’s correlation coefficient (r) between sensors ($L_{1}$ to $L_{6}$) for section-wise carving and parallel ski steering turns. Correlation values r > 0.50 are highlighted in bold, and those r > 0.70 are highlighted with the corresponding color of the turn phase.

		Carving	Parallel Ski Steering
			L_1_	L_2_	L_3_	L_4_	L_5_	L_6_	L_1_	L_2_	L_3_	L_4_	L_5_	L_6_
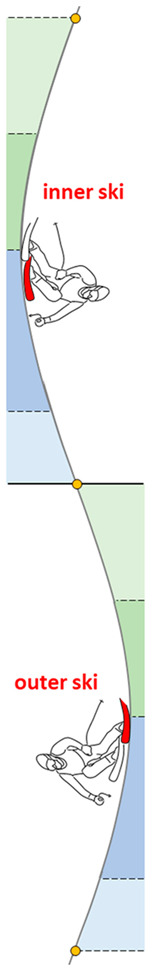	**Initiation**	L_1_	1						1					
L_2_	0.48	1					**0.77**	1				
L_3_	0.15	**0.88**	1				0.38	0.64	1			
L_4_	0.11	0.06	0.06	1			0.47	0.26	0.37	1		
L_5_	0.23	0.03	−0.02	**0.90**	1		**0.56**	0.34	0.24	**0.76**	1	
L_6_	0.35	−0.16	−0.27	**0.70**	**0.89**	1	**0.73**	0.48	0.26	**0.67**	**0.93**	1
**COM DC I**	L_1_	1						1					
L_2_	0.10	1					**0.67**	1				
L_3_	−0.08	**0.93**	1				0.18	**0.70**	1			
L_4_	0.21	−0.44	−0.41	1			0.05	0.13	0.25	1		
L_5_	0.22	−0.51	−0.52	**0.94**	1		0.23	0.29	0.23	**0.72**	1	
L_6_	0.19	−0.61	−0.63	**0.85**	**0.96**	1	0.37	0.35	0.15	0.38	**0.68**	1
**COM DC II**	L_1_	1						1					
L_2_	0.30	1					**0.70**	1				
L_3_	0.16	**0.94**	1				0.48	**0.78**	1			
L_4_	−0.18	**0.50**	**0.51**	1			0.42	**0.61**	0.47	1		
L_5_	−0.10	**0.54**	**0.57**	**0.96**	1		0.49	**0.53**	0.35	**0.91**	1	
L_6_	−0.12	**0.50**	**0.56**	**0.92**	**0.98**	1	**0.50**	0.30	0.13	**0.65**	**0.80**	1
**Completion**	L_1_	1						1					
L_2_	0.07	1					**0.79**	1				
L_3_	−0.10	**0.93**	1				**0.70**	**0.89**	1			
L_4_	0.08	0.05	0.01	1			0.21	0.22	0.18	1		
L_5_	0.18	−0.18	−0.23	**0.92**	1		0.17	0.18	0.15	**0.96**	1	
L_6_	−0.02	−0.24	−0.22	**0.75**	**0.88**	1	0.07	0.10	0.04	**0.86**	**0.90**	1
**Initiation**	L_1_	1						1					
L_2_	0.24	1					**0.67**	1				
L_3_	−0.21	**0.79**	1				0.35	**0.77**	1			
L_4_	0.10	0.01	−0.09	1			−0.03	−0.35	−0.34	1		
L_5_	0.19	−0.06	−0.20	**0.96**	1		0.12	−0.24	−0.31	**0.90**	1	
L_6_	0.20	−0.07	−0.21	**0.90**	**0.97**	1	0.23	−0.06	−0.14	**0.80**	**0.91**	1
**COM DC I**	L_1_	1						1					
L_2_	−0.07	1					**0.95**	1				
L_3_	−0.22	**0.94**	1				**0.73**	**0.81**	1			
L_4_	0.22	−0.10	−0.15	1			**0.66**	**0.67**	**0.70**	1		
L_5_	0.20	−0.07	−0.14	**0.94**	1		**0.76**	**0.79**	**0.76**	**0.96**	1	
L_6_	0.15	−0.04	−0.11	**0.80**	**0.93**	1	**0.82**	**0.85**	**0.75**	**0.87**	**0.95**	1
**COM DC II**	L_1_	1						1					
L_2_	**0.56**	1					**0.93**	1				
L_3_	0.48	**0.98**	1				**0.62**	**0.75**	1			
L_4_	0.28	**0.54**	**0.53**	1			**0.77**	**0.76**	**0.54**	1		
L_5_	0.33	**0.51**	**0.51**	**0.90**	1		**0.84**	**0.83**	**0.54**	**0.91**	1	
L_6_	0.24	0.23	0.24	**0.70**	**0.87**	1	**0.85**	**0.81**	0.48	**0.85**	**0.95**	1
**Completion**	L_1_	1						1					
L_2_	0.41	1					**0.89**	1				
L_3_	0.15	**0.89**	1				**0.68**	**0.83**	1			
L_4_	0.04	−0.11	0.02	1			**0.71**	**0.60**	0.44	1		
L_5_	0.16	0.02	0.13	**0.96**	1		**0.77**	**0.62**	0.37	**0.84**	1	
L_6_	0.22	−0.09	−0.03	**0.87**	**0.93**	1	**0.84**	**0.66**	0.45	**0.80**	**0.93**	1

## Data Availability

The data presented in this study are available on request from the corresponding author.

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
