# Peer review of "Technique-Dependent Relationship between Local Ski Bending Curvature, Roll Angle and Radial Force in Alpine Skiing"

_sensors, 2023, doi:10.3390/s23083997_

Round 1

Reviewer 1 Report

The authors have written a very detailed and comprehensive account of the basis of their work, the work itself and the established conclusions. The only thing which is lacking in the paper is the application of the found correlations ie a clear description of the ski design in the basis of the determined correlation. That is the only suggestion from this reviewer that is to add a section where an updated ski design be presented which will take in to account the newly established correlation. There are very minor English related typos which should also be taken cared off in the updated manuscript. 

Author Response

The authors have written a very detailed and comprehensive account of the basis of their work, the work itself and the established conclusions. The only thing which is lacking in the paper is the application of the found correlations ie a clear description of the ski design in the basis of the determined correlation. That is the only suggestion from this reviewer that is to add a section where an updated ski design be presented which will take in to account the newly established correlation. There are very minor English related typos which should also be taken cared off in the updated manuscript. 

Thank you for your time and effort in reviewing our work. Furthermore, thank you for the comments and suggestions that improve the paper. The authors have responded to your comments here and in the manuscript. All revisions to the manuscript have been marked with the "Track Changes" function so that they can be followed transparently.

The authors do not yet have the confidence to make any general statements or recommendations regarding ski designs, as these measurements were only performed with one test person. However, further measurements are planned to address this limitation. As you have correctly pointed out, information is missing in this regard in the manuscript, so the authors have added two sentences in the Conclusion.

Change in the paper at page 12 from line 427 to 431:

“To enhance ski design and achieve better equipment-athlete compatibility, future experimentation will involve testing different ski designs that vary in terms of bending and torsional stiffness, as well as incorporate left and right asymmetric designs. Furthermore, to enable more generalized performance statements for both the athletes and equipment, multiple athletes will be included in these tests.”

Reviewer 2 Report

This article has a high scientific quality and is of great importance for skiing. Defects are not recognizable.

Author Response

This article has a high scientific quality and is of great importance for skiing. Defects are not recognizable.

Thank you for your time and effort in reviewing our work. The authors are delighted to receive your feedback regarding the quality of their work, and are pleased to hear that you are satisfied. Thank you for your kind words!

For your information, minimal changes were made to the manuscript based on feedback from the other reviewers. All revisions to the manuscript have been marked with the "Track Changes" function so that they can be followed transparently.

Reviewer 3 Report

This is an interesting contribution to the existing literature, but the paper suffers from several shortcomings listed in the following comments.

-          The paper should be checked by a native.

-          A discussion section should be added.

-          The introduction should be updated by recent researches.

-          The novelty and contribution should be clearly bolded.

-          The authors should consider the following works:  

Transformer Winding Faults Detection Based on Time Series Analysis, IEEE Transactions on Instrumentation and Measurement 70, 1-10.

-          It’s better to suggest some subjects for future works.

Best regards,

Author Response

See attachment for response. 

Reviewer 4 Report

The paper presents an experimental investigation results on ski segmental deflection behavior by considering both carving and parallel ski steering in various turn phases. The obtained results seem reliable and accurate, they are deeply analyzed and clearly commented by using an excellent English style.

The following issues are recommended to improve the paper:

1.  Introduction: typically this section ends by stating the limits of the state of the art and the original contributions of the paper, along with the introduction of its Sections.

2. Recommendation to include a Nomenclature section for the used acronyms and symbols.

Author Response

Response see document in the attachment. 

Thank you very much.

Round 2

Reviewer 3 Report

Accept in present form.